# Sorption Constant of Bisphenol A and Octylphenol Onto Size-Fractioned Dissolved Organic Matter Using a Fluorescence Method

**DOI:** 10.3390/ijerph18031102

**Published:** 2021-01-27

**Authors:** Cheng-Wen Chuang, Wei-Shiang Huang, Hong-Sheng Chen, Liang-Fong Hsu, Yung-Yu Liu, Ting-Chien Chen

**Affiliations:** 1Department of Environmental Science and Engineering, National Pingtung University of Science and Technology, Pingtung 91201, Taiwan; a0921765948@yahoo.com.tw (C.-W.C.); stefsun921015@gmail.com (W.-S.H.); qmoro123@gmail.com (H.-S.C.); uny10331099@gmail.com (Y.-Y.L.); 2Department of Applied English, Tainan University of Technology, Tainan 71002, Taiwan; td0011@mail.tut.edu.tw

**Keywords:** bisphenol A, octylphenol, fluorescence quenching, dissolved organic matters, UV/Vis indicators

## Abstract

Dissolved organic matter (DOM) is a complex and heterogeneous mixture ubiquitously present in aquatic systems. DOM affects octylphenol (OP) and bisphenol A (BPA) distribution, transport, bioavailability, and toxicity. This study investigated OP and BPA sorption constants, log K_COC_, with three size-fractioned DOM. The molecular weights of the sized fractions were low molecular weight DOM (LDOM, <1 kDa), middle molecular weight DOM (MDOM, 1–10 kDa), and high molecular weight DOM (HDOM, 10 kDa–0.45 μm). The log K_COC_ ranged from 5.34 to 6.14 L/kg-C for OP and from 5.59 to 6.04 L/kg-C for BPA. The OP and BPA log K_COC_ values were insignificantly different (*p* = 0.37) and had a strong positive correlation (r = 0.85, *p* < 0.001). The OP and BPA LDOM log K_COC_ was significantly higher than the HDOM and MDOM log K_COC_ (*p* = 0.012 for BPA, *p* = 0.023 for OP). The average specific ultraviolet absorption (SUVA_254_) values were 32.0 ± 5.4, 13.8 ± 1.0, and 17.9 ± 2.8 L/mg-C/m for LDOM, MDOM, and HDOM, respectively. The log K_COC_ values for both OP and BPA had a moderately positive correlation with the SUVA_254_ values (r = 0.79–0.84, *p* < 0.002), which suggested the aromatic group content in the DOM had a positive impact on sorption behavior.

## 1. Introduction

Aquatic dissolved organic matter (DOM) is a complex mixture and consists of a wide variety of chemical compositions and molecular sizes as well as hydrophobic and hydrophilic components [1,2,3,4,5]. These properties affect various environmental behaviors of organic and inorganic contaminants, including toxicity, solubility, bioavailability, transport, and fate [6,7,8,9].

Previous studies have revealed that DOM physical-chemical properties are associated with the extent of interaction between DOM and hydrophobic organic compounds (HOC). For example, the DOM aromatic content, molecular weight, hydrophobicity, and humification degree are positively correlated with the extent of hydrophobic organic compounds (HOCs) binding [6,7,8,9,10,11,12,13,14].

Ultraviolet and visible (UV/Vis) spectroscopy can effectively analyze DOM properties. Spectral indicators have been defined, such as a specific ultraviolet absorption at 254 nm (SUVA_254_) [3,15]. The SUVA value shows a positive correlation with the DOM molecular weight, extent of humification, and content of aromatic groups [3,4,5,10,15].

Endocrine disrupting compounds that interfere with the endocrine systems of wild animals have been a concern worldwide [6,7,13,16]. Phenolic xenoestrogenic compounds (PXEC) such as octylphenol (OP) and bisphenol A (BPA) are moderately estrogenic compounds. They mimic natural hormones that bind to the estrogen receptors of organisms and interrupt the endocrine system of wildlife in the environment [17,18,19,20,21]. These compounds are heavily produced; the global annual production is more than a million metric tons for BPA [22] and a half million metric tons for alkylphenol [23]. They have various application functions; a high proportion of these compounds are released into the environment [22,23,24,25]. High concentrations in aquatic environments, even up to the hundreds of μg/L level, have been measured [26,27,28,29,30].

Since BPA and OP are moderately hydrophobic organic compounds, the octanol/water partition coefficient (log K_OW_) values are 3.32 and 4.12, respectively [24,29,30]. OP and BPA favor sorption onto hydrophobic DOM. The sorption of HOCs onto DOM is an important factor that can determine the transport, transformation, and fate of HOCs in aquatic systems [6,7,30]. Related studies had investigated PXEC sorption on colloids, such as commercial humic acid and fulvic acid as well as colloids isolated from wastewater biological treatment processes, river water, sewage effluent, and seawater [7,30,31,32,33]. These findings generally demonstrate that the partition coefficient normalized colloidal organic carbon (log K_COC_) values were greater than OP and BPA log K_OW_, and the values varied greatly from different sources. The different levels of humification and functional groups in DOM accounted for this discrepancy [7,30,31,32].

Fluorescence static quenching modeling (FSQM) [34] is an easily operated and sensitive method and is commonly used to study the sorption behavior of HOCs onto colloidal DOM and humic substances (HS), such as polycyclic aromatic hydrocarbons (PAHs) and estrogenic compounds [7,13,32,35,36,37]. BPA and OP sorption on environmental media such as soil, sediments, suspended particulate matter, and sludge [29,38,39,40] as well as colloids have been studied [7,14,30,33]. Yamamoto et al. [7] and Yeh et al. [13] used FSQM to study the sorption constants of OP and BPA on DOM and sediment humic substances. The log K_COC_ values ranged from 4.63 to 5.16 L/kg-C for OP and ranged from 4.80 to 5.09 L/kg-C for BPA.

The DOM in a water sample are operationally defined as a solids in the liquid phase passing through a filter (i.e., commonly <0.45 μm). Generally, the size of the DOM colloid is defined as a molecular weight higher than 1 kDa. However, the DOM in the filtrate contains various sized fractions, which may have different interactions with organic and inorganic compounds [14,30,33,41]. The sorption of BPA and OP onto different size fractioned DOM, especially the truly dissolved phase (<1 kDa), has had little attention. The truly dissolved DOM has been shown to have a strong sorption capacity to HOCs and heavy metals [42,43,44].

This study investigated the OP and BPA sorption constants on three size-fractioned groups of DOM collected from lake water. The filtered bulk DOM (BDOM) solution was sequentially separated into high molecular weight DOM (HDOM, 10 kDa–0.45 μm), middle molecular weight DOM (MDOM, 1–10 kDa), and low molecular weight DOM (LDOM, <1 kDa). The characteristics of the three DOM fractions were investigated with the UV/Vis aromaticity index SUVA_254_. Sorption constants between OP, BPA and sized DOM solutions were measured with a fluorescence static quenching modeling.

## 2. Methods and Materials

### 2.1. Water Sample Collection and Measurement

The water samples were collected from Lungluan Lake, which is located in a tropical area in the center of the famous National Kending Park, Taiwan. The lake is an important wetland and was designed as a National Wetland by the Ministry of the Interior, Taiwan. The lake area is about 1.75 km^2^, and the depth varies between 2–3 m. The lake’s major sources of water influx are rainfall and wastewater discharged from local residences. The lake water provides bird protection as well as a source of irrigation water for local farmlands. The sampling was performed at two sites shown in Appendix A: sampling map. The water samples were taken from below the water surface at 20–50 cm. At each sampling site, two water samples were taken at different sampling times. In total, the DOM separation process was performed on four water samples. In the formulation of the study, each 10 L representative water sample was retrieved from the subject lake and was stored in an iced refrigerator; the sample was then brought back to the laboratory within four hours. Subsequently, a 3 L volume of the water sample was passed through a 0.45 um filter (Pall) to generate the filtrate (BDOM). The filtrated BDOM was stored in a dark refrigerator for further analysis. Sodium azide (0.5%) was then added to the BDOM solution to inhibit the effect of bacteria on the DOM. The latter was stored at 4 °C in a refrigerator. To avoid contamination, the containers were previously washed with ultrapure water.

### 2.2. DOM Separation

In this study, a cross-flow ultrafiltration system sequence equipped with 10 and 1 kDa nominal molecular weight cutoff ceramic membrane cartridges was used (Filtanium, France). The separation process is shown in the (Appendix A: separation process and volumes of sized DOM). The feed flow rate was 1.7–2.0 L/min, and the membrane area was 320 cm^2^. The permeate flow rate was 12 and 25 mL/min for the 10 and 1 kDa cartridges, respectively, while the feed flow pressure was maintained at 5.0 and 1.5 kg/cm^2^, respectively. In each separation process, 3-L BDOM solutions were separated into the three size-fractioned DOM (10 kDa < HDOM < 0.45 μm, 1 kDa < MDOM < 10 kDa, and LDOM < 1 kDa). The concentration factor was kept at 10 (Equation (1)), in which the retentate flow was directed back to the feed flow bottle, and the permeate flow was directed to another container to be collected. The permeate volume was the feed flow for the second separation process. The flow volumes of feed, retentate, and permeate flow of the two separations are shown in Appendix A. The volumes of the separated size-fractioned DOM were 300, 270, and 2430 mL for HDOM, MDOM, and LDOM, respectively. The membrane system was cleaned and preconditioned before each macromolecule isolation experiment, as recommended by manufacture protocol. The three size-fractioned DOM and BDOM samples were measured for dissolved organic carbon (DOC) with UV-vis spectroscopy. The DOC concentrations of sized DOM were measured using a TOC analyzer (TOC-V CSH/CSN, Shimadzu, Japan).

The volume fraction (VF_i_) is the ratio of the permeate/retentate volume to the initial feed flow volume (3000 mL) for each size-fractioned DOM. The calculation of volume fraction has been added in Equation (2). The volume fractions for the study were 1.0, 0.1, 0.09, and 0. 81 for BDOM, HDOM, MDOM, and LDOM, respectively. The organic carbon (OC) mass balances (MB) were calculated by Equation (3). The OC mass fractions (MF_i_) for the size-fractioned DOM were calculated following Equation (4).
(1)$$C_{f}=\frac{V_{ret}+V_{perm}}{V_{ret}}$$
(2)$$VF_{i}=\frac{V_{i}}{V_{b}}$$
(3)$$\mathrm{MB}\;\left(\%\right)=\frac{\sum (C_{i}\times V_{i})}{C_{b}\times V_{b}}\times 100$$
(4)$$MF_{i}\;\left(\%\right)=\frac{C_{i}\times V_{i}}{\sum \left(C_{i}\times V_{i}\right)}\times 100$$
where *V_perm_* and *V_ret_* are the volumes of permeate and retentate in each separated step for *V_i_* and *V_i_*_+1_. *C_i_* and *V_i_* are the concentration and volume for each size-fractioned DOM. *C_b_* and *V_b_* are the concentration and volume for the feed flow.

### 2.3. UV/Vis Measurements

An aliquot of 1 mg-DOC/L from each of the four DOM solutions was measured using a UV/Vis spectrophotometer (Hitachi, U-2900, Tokyo, Japan) for absorbance measurement, on a scanning wavelength of 800–200 nm. Background was corrected according to the Helms et al. [1] method. An average absorbance value of 700–800 nm was adopted as the background value, which was subtracted from the sample value. The SUVA_254_ indicator followed Equation (5) and the SUVA_272_ and SUVA_280_ indicators were calculated in the same manner.
SUVA_254_ (L/m/mg-C) = (UV_254_/[DOC]) × 100(5)
where UV_254_ (cm^−1^) is the UV/Vis absorbance at 254 nm of the sample, and [DOC] is the DOC concentration (mg-C/L) of the size-fractioned DOM solutions [3,15]. Recoveries of the three SUVA indicators were determined by summation of the absorbance times, DOC concentration and VF_i_ of the three size-fractioned DOM solutions divided by the absorbance times of the DOC concentration in bulk solution. The calculation followed Equation (6).
R_SUVA_ (%) = Σ(UV_i_ × [DOC]_i_ × VF_i_)/(UV_b_ × [DOC]_b_ × VF_b_) × 100(6)

### 2.4. Fluorescence Spectroscopy

In this study, three-dimensional fluorescence excitation/emission matrix spectroscopy (EEM) was recorded. The emission wavelength (Em) was established as the EEM X-axis. The Y-axis was used as the excitation wavelength (Ex), and the Z-axis was the fluorescence intensity [2]. An aliquot of 0.5–5 mg-DOC/L, with and without OP/BPA for each of the sized DOM solutions, was measured by a fluorescence spectrophotometer (Hitachi, F-7000, Japan). Fluorescent scanning conditions were: an excitation wavelength of 200–450 nm with 5 nm increases, an emission wavelength of 250–550 nm with 2 nm increases, a scan rate of 2400 nm/min, a slit width of 5 nm, and voltage amplifying of 700 V. The spectra were obtained by subtracting an ultrapure water blank spectrum, recorded in the same condition, to eliminate the Raman scatter peaks. The fluorescence intensity of the DOM solution was detected at Ex/Em = 225/308 nm. The EEM plots of the standard solutions (OP, BPA) and the EEM plots of the OP and BPA interaction with DOM are shown in the (Appendix A: EEM plots of four sized DOM without OP and BPA at DOC 1 mg/L. Appendix A: EEM plots of OP/BPA interaction with size-fractioned DOM). The inner-filter correction follows the Holbrook et al. [6] procedure. The excitation/emission matrix (EEM) of the DOM solution at Ex/Em = 225/308 nm did not have any apparent fluorescence intensity for OP and BPA signals; therefore, the DOM containing low OP and BPA, and FSQM data, were ignored. The humification index (HIX) was calculated using the fluorescence data. HIX is the ratio of the area under the emission spectra over Em 435–480 nm to that over 300–345 nm at Ex 254 nm, which tends to increase with a higher degree of humification [2,4].

DOM, OP and BPA’s sorption constants were calculated using the fluorescence static quenching modeling (FSQM) (Equation (1)) [34,36]:(7)$$\frac{F_{0}}{F}=1+K_{COC}\;\left[DOC\right]$$
where *F* and *F*_0_ are the fluorescence intensity of the standard OP and BPA solution with and without DOM solution present. [DOC] is the DOM concentration (mg-C/L) and *K_COC_* is the sorption constant (L/mg-C). It should be noted that Equation (7) assumes the PXEC sorption on the DOM solution had static quenching [36]. The fluorescence static quenching modeling was conducted on 24 samples that included both OP and BPA with three size-fractioned DOM of the four sampling waters. The 24 FSQM tests were fitted with fluorescence static quenching modeling. The K_COC_ was determined through the linear regression of the F_0_/F values with the [DOC] concentration. The linear significance was dependent on whether the slope and intercept had the necessary significance (*p* < 0.05) to determine the suitability of the fluorescence static quenching modeling. The linear slope was the *K_COC_* (L/mg-C).

### 2.5. Statistical Analysis and Calculation of Fluorescence Data

In this study, S-Plus software (V 6.2, Palo Alto, CA, USA) was used to determine the UV-Vis indicator and sorption constant differences and to perform linear regression. The indicator and sorption constant difference test for the three size-fractioned DOM solutions used ANOVA test methods at significance levels (*p* < 0.05).

## 3. Results and Discussion

### 3.1. DOC Concentration and Carbon Mass Fraction

Generally, DOM filtrate passed through 0.45 μm is used to represent DOC concentration [3]. Table 1 lists each size-fractioned DOC concentration for average DOC in the samples. The DOC concentrations for the filtrate (BDOM) ranged from 2.67 to 3.02 mg-C/L.

In other work, the DOC concentrations in Lake Biwa, Japan, ranged from 1.06 to 1.28 mg-C/L [45], and DOC concentrations were between 2.25 and 2.89 mg/L in Lake Hongfeng and Lake Baihua, China [46]. Helms et al. [1] measured wetland and oceanic water DOC concentrations where average DOC concentrations ranged from 1.94 to 15.35 mg/L for four sites. Town and Filella [47] classified low DOC concentration in lake water as less than 7 mg-C/L, which suggested the DOC concentration in this studied site was in the low concentration range.

The carbon mass balances (Equation (3)) for the four samples ranged from 84 to 103%, and the average mass balance was 91.6%, which was in a reasonable range when compared to the literature values ranging from 80 to 120% [30,33,45,48,49]. For example, Wu and Tanoue [45] separated lake water into three size-fractioned DOMs that included 0.1–0.7 μm, 5 kDa–0.1 μm, and <5 kDa fractions. The carbon mass balances were 89–109%. The river water was separated into colloidal fractions (1 kDa–0.45 μm) and truly dissolved (<1 kDa) phases, where the organic carbon mass balances were 108% and 87%, respectively [30,33]. The organic carbon mass balances were within reasonable range, which suggested that OC loss was low. It implied the separation system is acceptable for conducting FSQM experiments.

The carbon mass fractions (Equation (4)) in this study were 14.8 ± 1.3%, 17.4 ± 1.3%, and 67.8 ± 1.9% for HDOM, MDOM, and LDOM fractions, respectively. Town and Filella [47] reported that carbon mass fractions were 33% and 67% for colloidal (>1 kDa) and truly dissolved fractions (<1 kDa), respectively, for the low DOC concentration profile (<7 mg-C/L) in lake water. In the Lake Biwa, the three size-fractioned DOM carbon masses were 54–69% (<5 kDa), 30–34% (5 kDa–0.1 μm), and 1–2% (0.1–0.7 μm) [45]. Zhou et al. [49] reported the colloidal carbon mass fractions were 57–72% and 43–68% for river and seawater, respectively. Wen et al. [48] separated sea water into colloidal (1 kDa–0.45 μm) and truly dissolved phases (<1 kDa), in which the carbon mass fractions were 45% and 55% for colloidal and truly dissolved phases, respectively. The DOM has varied properties and colloidal carbon mass fractions as well [45]. In this study, carbon mass fractions were 32 and 68% for colloidal (1 kDa–0.45 μm) and truly dissolved (<1 kDa) phases, respectively. These are similar to the carbon mass fractions measured in freshwater [45,47]; the colloidal phase is less than the fraction surveyed in river and seawater [48,49]. The OC mass fractions of LDOM averaged 68%. The percentage and log K_COC_ of LDOM were higher than H/MDOM (Section 3.2) in this study, which needs attention. The LDOM may play an important role in HOC’s sorption.

### 3.2. UV-Vis Absorption Spectroscopy of DOM

UV-Vis wavelength absorbance between 220 and 280 nm is considered suitable for reflecting the properties of natural organic matter [1,3,4,50]. The average UV-Vis absorbance wavelengths between 235 and 300 nm for the four DOM solutions are shown in Figure 1. The absorbance steeply increases at wavelengths less than 260 nm and flattens at wavelengths greater than 260 nm. The UV-Vis absorbance trend is similar to absorbance measured with river water and water-extracted organic matter from sediment and compost material [13,44,51,52]. The BDOM is separated into size-fractioned HDOM, MDOM, and LDOM. The size-fractioned DOMs have the same origin: absorbance wavelengths between 235 and 300 nm respond to the content of aromaticity in size-fractionated DOM.

In this study, three UV-Vis specific ultraviolet visible absorbance values were calculated (SUVA_254_, SUVA_272_, and SUVA_280_), which are listed in Table 1. The recoveries of the three indicators (Equation (6)) for the three sized fractions to bulk solution ranged from 99 to 108%. The average fraction ranges were 75–81%, 9–13%, and 10–13% for LDOM, MDOM, and HDOM solutions. LDOM had a much higher fraction than HDOM and MDOM, both in terms of carbon mass and UV absorbance. The UV-Vis indicator recovery was comparable to the UV-Vis recoveries for absorbance at 254 nm, ranging from 84% to 136% [44,45].

Three SUVA values are widely applied to characterize DOM properties; the values had a strong positive correlation with DOM aromatic content. A higher DOM SUVA value indicates more humic substances and a higher extent of humification [10,15]. The LDOM SUVA_254_ and SUVA_272_ values were significantly higher than the values of MDOM (*p* = 0.013 and 0.044, respectively), but the SUVA_280_ values of the three sized DOM solutions were not significantly different (*p* = 0.26). In addition, the three SUVA indicators (254, 272, and 280 nm) had positive correlations among the three-sized DOM solutions (r = 0.83–0.98, *p* < 0.001), which suggested the three SUVA indicators could reflect the aromatic content of DOM. Nevertheless, the correlations between the three SUVA indicators and log K_COC_ values showed that SUVA_254_ had a better correlation than SUVA_272_ and SUVA_280_. Therefore, SUVA_254_ was used to examine the correlation with log K_COC_ in this study.

Matilainen et al. [3] and Hansen et al. [4] reported that when water samples’ SUVA_254_ values are greater than 4.0, the major composition of DOM primarily contains hydrophobic substances. When the values are less than 3.0, the DOM mainly contains hydrophilic substances. In this study, the average SUVA_254_ values ranged from 13.8 to 32.0, which suggested the DOM primarily contained hydrophobic substances and specific aromatic compounds.

The composition of aquatic DOM had two major sources including terrestrial and aquatic sources [47]. The terrestrial sources resulted from soil washout, generally comprised mostly with a refractory fulvic acid-like substance. Aaquatic sources were comprised of substances excreted by aquatic biota and their degradation products. Aquatic DOM commonly had a larger size molecular weight than the terrestrial source. In this study, LDOM may have resulted from soil washout in a nearby area. The EEM peak of the sized DOM (Appendix A) is centered at wavelength Ex/Em 225/390–410 nm, which indicates that the major component of the sized DOM is a fulvic acid-like substance [2]. The HIX values (Table 1) suggested low humification of the sized DOM [2,4].

### 3.3. Sorption Constants between Size-Fractioned DOM, and OP and BPA

OP and BPA sorption constants in size-fractioned DOM solutions were measured with a fluorescence static quenching model. The K_COC_ value was obtained from linear regression of the relationship between the ratio of the fluorescence intensity ratios (F_0_/F) and the DOC concentrations using Equation (7). Figure 2a–c are the ratios F_0_/F of OP and BPA to the represented LDOM, MDOM, and HDOM concentrations, respectively, and the regressions curves. The F_0_/F increased linearly, following the increased DOC concentrations. This indicated that intensity (F) decreased following the increase of DOC concentrations. The sorption of OP and BPA reduced the standard OP and BPA solutions’ fluorescence intensity onto the DOM. Each showed a significant linear relationship; the slopes were the K_COC_ values. LDOM, MDOM, and HDOM had similar curves, but the slopes were different. In practice, sorption constants can be obtained by conducting FSQM. In the present study, LDOM, MODM, and HDOM had the same origin DOM but different aromaticity content, which resulted in different sorption constants. The average and standard deviations of log K_COC_ values are listed in Table 2. The average intercepts of regression were 0.69–1.14, and r^2^ was 0.86–0.95. This suggested static fluorescence was the main mechanism, and the fluorescence static quenching modeling was approved to calculate the K_COC_ values. The FSQM method had been used to test sorption constants between colloids and several intermediate hydrophobic organic compounds, including estrone (E1), 17β-estradiol (E2), estriol (E3), 17α-ethynylestradiol (EE2), nonylphenol (NP), OP, and BPA, where the intercept ranged from 0.60 to 1.18, and r^2^ ranged from 0.55 to 0.98 [6,7,9,13,53].

The log K_COC_ values of BPA and OP sorption onto size-fractioned DOM ranged from 5.34 to 6.14 for BPA and from 5.59 to 6.04 for OP. OP and BPA had strong sorption capacity onto these DOM fractions. The log K_COC_ values showed no significant difference between OP and BPA sorption onto DOM (*p* = 0.37), and the log K_COC_ values had a positive linear relationship as shown in Figure 3 (r = 0.85, *p* < 0.001). In each size-fractioned DOM, the log K_COC_ values were not significantly different between OP and BPA (*p* = 0.05–0.42).

When comparing the log K_COC_ values of BPA and OP in three size-fractioned DOM, the log K_COC_ values of OP sorption onto LDOM were significantly greater than sorption onto HDOM (*p* = 0.023), and the log K_COC_ values of BPA sorption onto LDOM were significantly greater than the sorption onto MDOM (*p* = 0.012). Both OP and BPA had a high sorption capacity onto the LDOM, which is in contrast to previous studies of estrogen and PXEC sorption onto size-fractioned DOM [13,31,54,55]. The HOCs had a higher sorption capacity onto the higher molecular weight DOM than the low molecular weight DOM, owing to the higher aromatic content and extent of humification [13,31,54,55]. However, previous and present studies have shown that LDOM had higher sorption constants than HDOM. The sorption constants between herbicide prometryne and lakebed sludge DOM were 3.55, 3.38, and 2.74 for molecular weight <3.5 kDa, 3.5–14 kDa, and >14 kDa, respectively [43]. McPhedran et al. [42] have tested partitioning of HOCs 1,2,4,5-tetrachlorobenzene, pentachlorobenzene, and hexachlorobenzene to DOM for both 1.5 μm and 1 kDa filtrates of primary effluent from a municipal wastewater treatment plant using the gas-stripping technique. The partitioning to DOM < 1 kDa dominated the overall partitioning of the three chlorobenzenes in the 1.5 μm filtrate. They indicated that significant partitioning of HOC may occur to DOM < 1 kDa and highlighted the need for further experiments with other HOCs and DOM characterization to better understand and explain the observed partitioning.

Several studies have been conducted involving OP and BPA sorption experiments onto solid phases. In this regard, such sorption experiments were performed on unaltered soil, sediment, and suspended particulate matter, as well as sorption on humic substances extracted from sediment and isolated from water DOM [6,7,29,30,38,39,40]. The organic carbon-normalized sorption constants log K_OC_ for solid liquid partition and partition constant log K_COC_ between colloidal and truly dissolved phases of OP and BPA, which had a large variation. In this study, the total sorption constants were 5.79 ± 0.14 and 5.71 ± 0.26 for OP and BPA, respectively. The log K_COC_ values were one to three orders higher than the OP/BPA sorption constants log K_OC_ between solid/liquid phases and the sorption constants log K_COC_ between colloidal (>1 kDa) and truly dissolved (<1 kDa) phases [7,13,16,29,31,32,56,57,58,59]. The sorption constants log K_COC_ were comparable to log K_COC_ between colloidal and truly dissolved phases [30] but one order lower than log K_COC_ between colloidal and truly dissolved phases [14,33].

In this study, the sorption constants log K_COC_ showed that LDOM had higher log K_COC_ values than MDOM and HDOM. The log K_COC_ values are commonly greater than OP and BPA sorption onto soil, sediment, suspended particulate matter, and colloidal DOM. HIX was insignificantly correlated with the log K_COC_ for both OP and BPA (r = −0.46, *p* = 0.14 for OP; r = −0.49 *p* = 0.10 for BPA). K_COC_ was strongly affected by the aromatic properties of DOM but unaffected by the humification degree of DOM, in keeping with the sorption behavior between DOM and HOC reported by Hur et al. [60]. High log K_COC_ values in the truly dissolved phase may be attributed to high aromaticity and low humification. Previous studies [6,7,31,32,42] hypothesized that DOM < 1 kDa insignificantly contributed to hydrophobic organic compound partitioning. This limit has only been verified experimentally for a few sorbate/sorbent systems [42,43,44]. In our study, the sorption constants between OP/BPA and size-fractioned aquatic DOM using the FSQM method showed that LDOM had a higher sorption constant than M/HDOM. Moreover, the organic carbon mass fraction was 68% for LDOM. The high sorption constant between LDOM and OP/BPA may have had a strong effect on OP and BPA fate, transport, and biotoxicity in the aquatic environment. Our and McPhedran et al.’s [42] results are in contrast to previous hypothesis that may use more experiments to test truly dissolved DOM (<1 kDa) sorption with various hydrophobic organic compounds at various environmental conditions.

### 3.4. Correlation between UV-Vis Indicators and Log K_COC_

HOC’s sorption onto DOM was affected by HOC’s hydrophobicity, the DOM’s chemical properties, and environmental conditions. In this study, OP and BPA had significant sorption capacity onto three size-fractioned DOM. The log K_OW_ values of BPA and OP were 3.32 and 4.12, respectively, which have approximately one order of difference. However, the log K_COC_ values of OP and BPA sorption onto DOM showed no significant difference, which suggested the hydrophobic property of BPA and OP may not be the dominant factor that controls sorption onto DOM. This is in contrast to some studies that investigated with unaltered environmental media such as soil, sediment, and suspended particulate matter [29,39,61]. However, log K_COC_ was found to be independent from log K_OW_. This has also been observed in estrogens and phenolic estrogens sorption onto colloidal DOM. Holbrook et al. [6] and Yamamoto et al. [7] suggested nonspecific hydrophobic sorption may not be the major factor that controls sorption.

In this study, OP and BPA had higher sorption constants onto LDOM than sorption constants onto HDOM and MDOM, and values of SUVA_254_ of LDOM were greater than HDOM and MDOM. Figure 4 show the linear correlation between log K_COC_ and SUVA_254_, which has a medium positive correlation with sorption constant log K_COC_ for OP and BPA (r = 0.81–0.82, *p* < 0.002). This suggests DOM’s chemical structure and composition have a positive impact on BPA and OP sorption on DOM. The SUVA_254_ indicator is a surrogate for the abundance of aromatic functional groups in DOM. A greater amount of aromatic content functional groups increased the sorption constant, which demonstrated that aromatic functional groups had a positive effect on sorption capacity.

It has been observed in other studies that the sorption constants of estrogens and phenolic estrogens had a positive correlation with UV-Vis absorbance at 254, 272, and 280 nm [6,7,8,32]. Values of indicators SUVA_254_ (272 and 280) have the similar property of hydrogen bond interaction [6,7,13,31,32]. Log K_COC_ with a strong affinity sorption on sized DOM can be attributed to colloidal DOM containing high amounts of aromatic functional groups. These results suggested the π−π* electron of HOC interaction with DOM is an important mechanism, giving rise to the sorption of BPA and OP on DOM [6,7,32].

## 4. Conclusions

In the present study, we hypothesized that size-fractioned DOM had different PXEC sorption capacity, and chemical properties influenced the sorption constants. The sorption constants of OP and BPA on size-fractioned DOM isolated from lake water were obtained using the FSQM method. LDOM had a higher carbon mass fraction and sorption capacity than HDOM and MDOM. The UV-Vis aromaticity indicator (SUVA_254_) suggested the LDOM had higher aromatic functional groups in their structure than MDOM and HDOM. SUVA_254_ had a positive correlation with sorption constant log K_COC_. The extent of aromatic functional groups in the DOM samples was the main factor that influenced sorption ability. Previous studies hypothesized that DOM < 1 kDa insignificantly contributed to hydrophobic organic compound partitioning [6,7,31,32,42]. The present study results suggested that the sorption behavior and chemical properties in low molecular weight DOM should be considered when investigating hydrophobic organic compound sorption on DOM. The fluorescence method is an effective method for obtaining the sorption constant and chemical properties of DOM. DOM separation into size-fractioned DOM to investigate the sorption constant of different molecular weight DOM is essential to understand BPA/OP’s sorption behavior.

## Figures and Tables

**Figure 1 ijerph-18-01102-f001:**
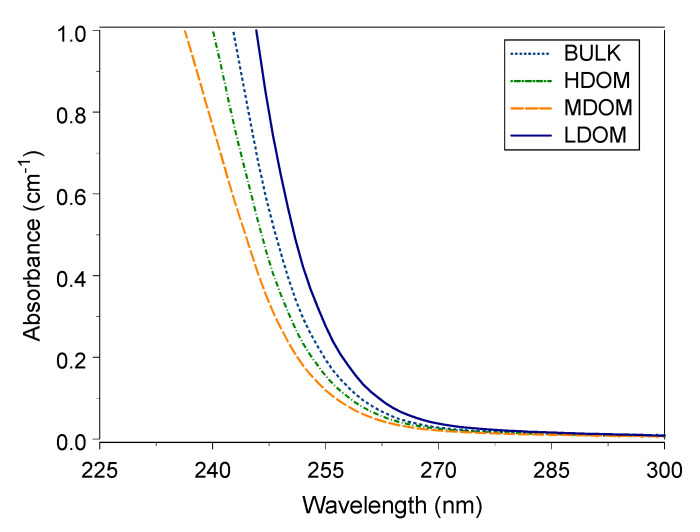
UV-Vis absorbance curve of bulk and size-fractioned DOM.

**Figure 2 ijerph-18-01102-f002:**
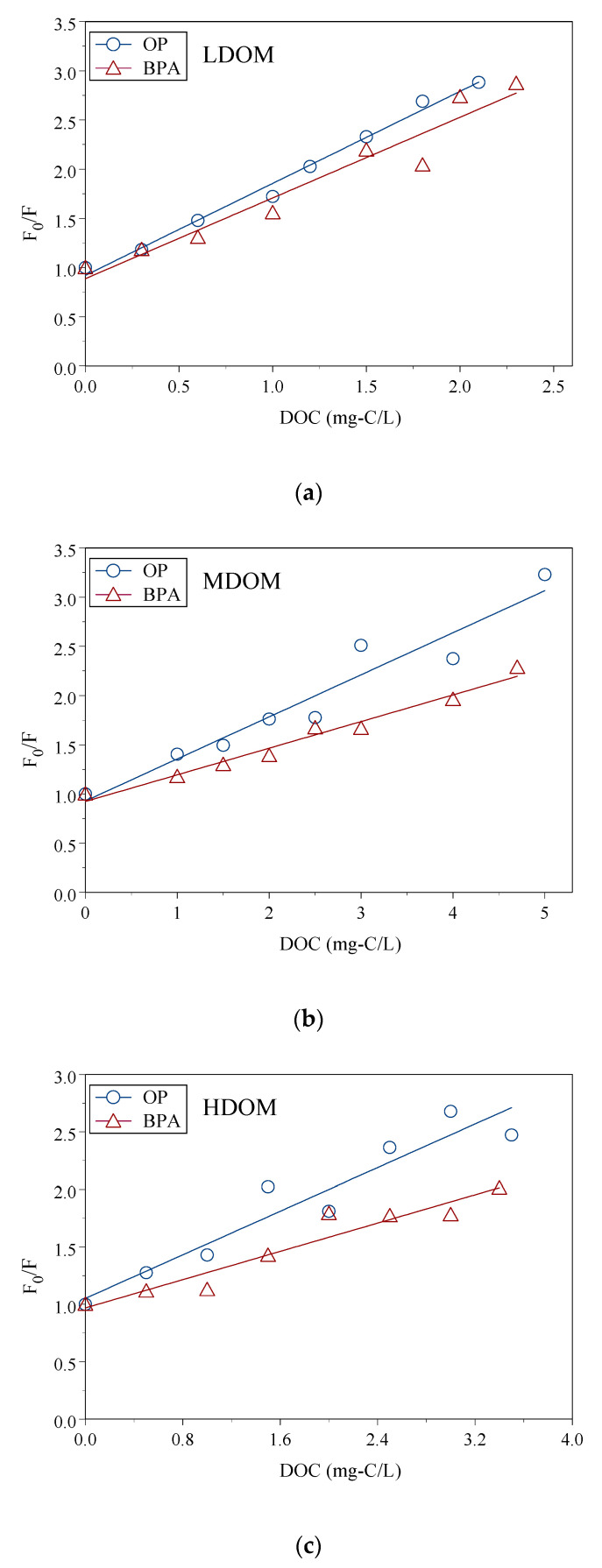
The linear regression between ratios of F_0_/F for OP/BPA and DOC concentrations of size-fractioned DOM: LDOM (**a**), MDOM (**b**), and HDOM (**c**).

**Figure 3 ijerph-18-01102-f003:**
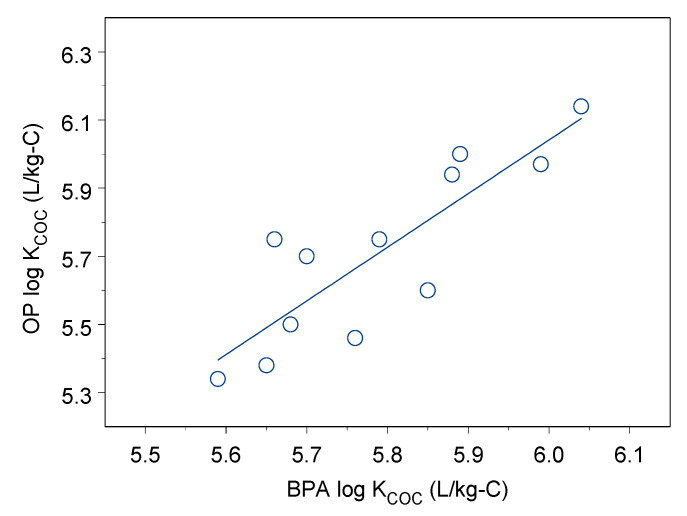
Linear regression between OP and BPA log K_COC_.

**Figure 4 ijerph-18-01102-f004:**
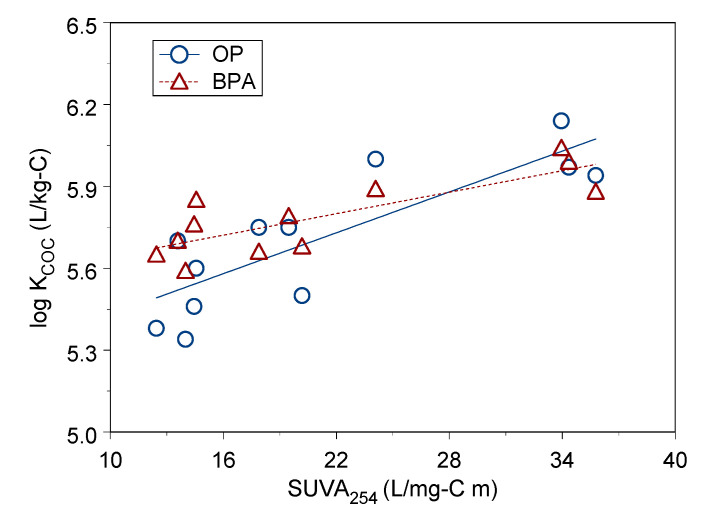
Linear correlation between log K_COC_ and SUVA_254_ for OP and BPA.

**Table 1 ijerph-18-01102-t001:** Dissolved organic carbon (DOC) concentrations and UV/vis and fluorescence indicators of size-fractioned dissolved organic matter (DOM) and BDOM (*n* = 4).

Samples	DOC (mg-C/L)	SUVA_254_	SUVA_272_	SUVA_280_	HIX
BDOM	2.89 ± 0.16	22.5 ± 1.8	2.40 ± 0.28	1.55 ± 0.22	2.43 ± 0.30
HDOM	3.93 ± 0.64	17.9 ± 2.8	2.18 ± 0.53	1.48 ± 0.43	2.28 ± 0.13
MDOM	5.10 ± 0.30	13.8 ± 1.0	1.81 ± 0.14	1.23 ± 0.10	2.72 ± 0.21
LDOM	2.23 ± 0.32	32.0 ± 5.4	3.18 ± 0.81	1.99 ± 0.63	1.95 ± 0.51

**Table 2 ijerph-18-01102-t002:** Summary of linear regression between octylphenol (OP) and bisphenol A (BPA) and size-fractioned DOM using the fluorescence static quenching modeling.

Compounds	Size Fraction	Log K_COC_	y-Intercept	R^2^
OP	>10 kDa	5.68 ± 0.08	1.14 ± 0.30	0.86 ± 0.06
	1–10 kDa	5.74 ± 0.09	0.69 ± 0.25	0.93 ± 0.02
	<1 kDa	5.95 ± 0.08	0.95 ± 0.10	0.95 ± 0.05
BPA	>10 kDa	5.58 ± 0.20	0.80 ± 0.18	0.93 ± 0.04
	1–10 kDa	5.53 ± 0.14	0.78 ± 0.11	0.95 ± 0.03
	<1 kDa	6.01 ± 0.09	0.99 ± 0.31	0.95 ± 0.04

## Data Availability

Not applicable.

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
