# Peer review of "Sorption Constant of Bisphenol A and Octylphenol Onto Size-Fractioned Dissolved Organic Matter Using a Fluorescence Method"

_ijerph, 2021, doi:10.3390/ijerph18031102_

Round 1

Reviewer 1 Report

The experimental work of the paper was carried out in detail, and the experimental data obtained was detailed and reliable. The experimental plan is designed reasonably. However, there are relatively few introductions about other people’s research work in this field, and at the same time, there is no comparison with other people’s research results in this field.

Author Response

Comments and Suggestions for Authors

The experimental work of the paper was carried out in detail, and the experimental data obtained was detailed and reliable. The experimental plan is designed reasonably. However, there are relatively few introductions about other people’s research work in this field, and at the same time, there is no comparison with other people’s research results in this field.

Response #1

Thank you for your comment.

In the Introduction Section Lines 67-70, we have added OP and BPA's sorption constants conducted on FSQM reported on literature.

Lines 67-70: Yamamoto et al. [7] and Yeh et al. [13] used FSQM to study sorption constants of OP and BPA on DOM and sediment humic substances. The log KCOC values ranged from 4.63 to 5.16 L/kg-C for OP and ranged from 4.80 to 5.09 L/kg-C for BPA.

The reported sorption constants of OP and BPA including DOM, soil, sediment, suspended particulate matter, and humic substances. Lines 305 to 315 discussed the sorption constants obtained in this study compared to various media reported values in literature.

Lines: 305 – 315. Several studies have been conducted involving OP and BPA sorption experiments onto solid phases. In this regard, such sorption experiments were performed on unaltered soil, sediment, and suspended particulate matter as well as sorption on humic substances extracted from sediment and isolated from water DOM [6,7,29,30,38-40]. The organic carbon normalized sorption constants log KOC for solid liquid partition and partition constant log KCOC between colloidal and truly dissolved phases of OP and BPA had a large variation. In this study, the total sorption constants were 5.79±0.14 and 5.71±0.26 for OP and BPA, respectively. The log KCOC values were one to three orders higher than the OP/BPA sorption constants log KOC between solid/liquid phases and the sorption constants log KCOC between colloidal (> 1 kDa) and truly dissolved (< 1 kDa) phases [7,13,16,29,31,32,56-59]. The sorption constants log KCOC were comparable to log KCOC between colloidal and truly dissolved phases [30] but one order lower than log KCOC between colloidal and truly dissolved phases [14,33].

Reviewer 2 Report

The manuscript entitled: 'Sorption Constant of Bisphenol A and Octylphenol onto Size-fractioned Dissolved Organic Matter Using a Fluorescence Method' is well written and clearly shows its results. The results are interesting and may help the researchers in their work. The manuscript should be considered for publication.

Author Response

Comments and Suggestions for Authors

The manuscript entitled: 'Sorption Constant of Bisphenol A and Octylphenol onto Size-fractioned Dissolved Organic Matter Using a Fluorescence Method' is well written and clearly shows its results. The results are interesting and may help the researchers in their work. The manuscript should be considered for publication.

Response #2

Thank you for your comment.

Reviewer 3 Report

Dear Authors,

Please explain in the text the physical meaning of "mass balances more than 100%".

What is the factor of temperature on your measurement, mainly on fluorescence spectra? Explain in the text.

Haw you can explain the similar trend of UV and VIS absorbance curves at Figure 1? Explain in the text.

Your fluorescence static quenching models have similar curves for LDOM, MDOM and HDOM. How you can in practice distinguish which model should be use? Do you have some other identification methods? Explain in the text.

Please rewrite chapter "4. Conclusion" so that you clearly state your research intentions, the obtained results and explain the general relevance of your research.

Define the abbreviations list at the end of your manuscript all abbreviations used in the text.

Author Response

Comments and Suggestions for Authors

Comment #3-1

The physical meaning of "mass balances more than 100%".

Response #3-1

Thank you for your comment and suggestion.

The mass balances of more than 100% suggest the cartridge of the separation process is contaminated. Several reasons may cause the result. The residual organic matter in the cartridge is released in the next separation. It may be the measurement uncertainty of DOC concentration in the bulk and size-fractioned DOM.

Comment #3-2

What is the factor of temperature on your measurement, mainly on fluorescence spectra?

Response #3-2

Thank you for your comment.

We agree with the reviewer's comment. The temperature is one of the important parameters on fluorescence spectra. In our study, the temperature is kept at room temperature; hence, the temperature is not variable on the current study's fluorescence spectra.

Comment #3-3

Haw you can explain the similar trend of UV and VIS absorbance curves at Figure 1? Explain in the text.

Response #3-3

Thank you for your comment and suggestion.

In Line 221-223. We explain the similar trend of UV-Vis absorbance curves in Figure 1.

In Figure 1. The UV-Vis absorbance trend is similar to absorbance measured with river water and water extracted organic matter from sediment and compost material [13,44,51,52]. The BDOM is separated into size-fractioned HDOM, MDOM, and LDOM. The size-fractioned DOMs have the same origin: the absorbance wavelengths between 235 and 300 nm respond to the content of aromaticity in size-fractionated DOM.

Comment #3-4

Your fluorescence static quenching models have similar curves for LDOM, MDOM and HDOM. How you can in practice distinguish which model should be use? Do you have some other identification methods? Explain in the text.

Response #3-4

Thank you for your comment and suggestion.

The fluorescence static quenching model (FSQM) is intended to obtain the sorption constant between fluorescence organic compound (FOC) and DOM. The model assumes the sorption between FOC and DOM decreasing FOC fluorescence intensity and the ratio of F0/F and DOC concentration is a linear relationship. The slope of the linear relationship represents the sorption constant. The principal and results of the present study mention in Lines 261-166. Lines 265 – 268 comment in practice, how to distinguish which model should be used.

Lines 261 to 266. Figure 2a -2c are the ratios F0/F of OP and BPA to the represented LDOM, MDOM, and HDOM concentrations, respectively, and the regressions curves. The F0/F increased linearly, following the increased DOC concentrations. This indicated the intensity (F) decreased following the increase of DOC concentrations. The sorption of OP and BPA reduced the standard OP and BPA solutions' fluorescence intensity onto the DOM. Each showed a significant linear relationship; the slopes were the KCOC values. LDOM, MDOM, and HDOM had similar curves, but the slopes are different. In practice, the sorption constants can be obtained by conducting FSQM. In the present study, the LDOM, MODM, and HDOM had the same origin DOM but had different aromaticity content that resulted in the different sorption constants. Besides, this study found the dominant factor to control sorption constant is DOM aromaticity. The aromaticity is a parameter to distinguish the difference of sorption constant among size-fractioned DOM.

Comment #3-5

Please rewrite chapter "4. Conclusion" so that you clearly state your research intentions, the obtained results and explain the general relevance of your research.

Response #3-5

Thank you for your comment and suggestion.

The Conclusion Section has rewrite.

In the present study, we hypothesized size-fractioned DOM had different PXEC sorption capacity, and the chemical properties influenced the sorption constants. The sorption constants of OP and BPA on size-fractioned DOM isolated from lake water were obtained using the FSQM method. The LDOM had a higher carbon mass fraction and sorption capacity than the HDOM and MDOM. The UV-Vis aromaticity indicator (SUVA254) suggested the LDOM had higher aromatic functional groups in their structure than the MDOM and HDOM. The SUVA254 had a positive correlation with sorption constant log KCOC. The extent of aromatic functional groups in the DOM samples was the main factor that influenced sorption ability. Previous studies hypothesized that the DOM < 1 kDa insignificantly contributed to the hydrophobic organic compound partitioning [6,7,31,32,42]. The present study results suggested the sorption behavior and chemical properties in low molecular weight DOM should consider in investigating hydrophobic organic compound sorption on DOM. The fluorescence method is an effective method that the obtaining sorption constant and chemical properties of DOM. The DOM separation into size-fractioned DOM to investigate sorption constant of different molecular weight DOM is essential to understand BPA/OP's sorption behavior.

Comment #3-6

Define the abbreviations list at the end of your manuscript all abbreviations used in the text.

Response #3-6

Thank you for your suggestion.

Lines 378-380. We have listed the abbreviations used in the text.

List of abbreviation

DOC

dissolved organic carbon

DOM

dissolved organic matter

OP

octylphenol

BPA

bisphenol A

log KCOC

partition coefficient normalized colloidal organic carbon

BDOM

the filtered bulk DOM solution (< 0.45 μm)

LDOM

low molecular weight DOM (< 1 kDa)

MDOM

middle molecular weight DOM (1-10 kDa)

HDOM

high molecular weight DOM (10 kDa-0.45 μm)

SUVA254

specific ultraviolet absorption at 254 nm

HOC

hydrophobic organic compound

UV/Vis

ultraviolet and visible spectroscopy

PXEC

phenolic xenoestrogenic compounds

log KOW

octanol/water partition coefficient

FSQM

fluorescence static quenching modeling

HS

humic substances

PAHs

polycyclic aromatic hydrocarbons

EEM

three-dimensional fluorescence excitation/emission matrix

HIX

humification index

E1

estrone

E2

17β-estradiol

E3

estriol

EE2

17α-ethynylestradiol

NP

nonylphenol

log KOC

the OC normalized sorption constants for solid-liquid partition

VFi

the volume fraction

MB

organic carbon mass balances

MFi

organic carbon mass fractions

Reviewer 4 Report

This is a well planned and executed study. Presentation is clear and findings appear of value to the research community on the role of soluble organic compounds on the mobility of model endocrine disrupting compounds in the environment.

you have presented valid DOM separation and characterisation methods and data and have shown components of humic and fulvic like subsatnces extracted from real environmental media rather than purified reagent grade in laboratory experiments. this is to be supported. I have a few minor ocmments which would require some modificaiton of the text/wider discussion.

  1. only one source of DOM used in this experiment. this is obviously a limitation but is acceptable given depth of study on other aspects. Please expand discussion of implications of this for wider sources of DOM - particularly where these compounds are more likely to be at higher levels - e.g. wastewater
  2. solution composition. there is plenty of evidence to show impacts of other water quality paramters on the nature of DOM and also interaction during the sorption of organic compounds to DOM/OM. salinity, other ions (complexing/competing), pH etc. you have only one set of experimental conditions, these aren ot fully described, but the dynamic external environment should be ocnsidered
  3. above points will also impact on measurement conditions for fluorescence and UV VIS. commentary on this also needed.

Author Response

Comment #4

Comments and Suggestions for Authors

This is a well planned and executed study. Presentation is clear and findings appear of value to the research community on the role of soluble organic compounds on the mobility of model endocrine disrupting compounds in the environment.

You have presented valid DOM separation and characterisation methods and data and have shown components of humic and fulvic like substances extracted from real environmental media rather than purified reagent grade in laboratory experiments. this is to be supported.

Response #4

Thank you for your comments.

Comment #4-1

I have a few minor comments which would require some modification of the text/wider discussion.

Only one source of DOM used in this experiment. this is obviously a limitation but is acceptable given depth of study on other aspects. Please expand discussion of implications of this for wider sources of DOM - particularly where these compounds are more likely to be at higher levels - e.g. wastewater

Response #4-1

Thank you for your comment and suggestion.

The Conclusion Section has rewrite.

In the present study, we hypothesized size-fractioned DOM had different PXEC sorption capacity, and the chemical properties influenced the sorption constants. The sorption constants of OP and BPA on size-fractioned DOM isolated from lake water were obtained using the FSQM method. The LDOM had a higher carbon mass fraction and sorption capacity than the HDOM and MDOM. The UV-Vis aromaticity indicator (SUVA254) suggested the LDOM had higher aromatic functional groups in their structure than the MDOM and HDOM. The SUVA254 had a positive correlation with sorption constant log KCOC. The extent of aromatic functional groups in the DOM samples was the main factor that influenced sorption ability. Previous studies hypothesized that the DOM < 1 kDa insignificantly contributed to the hydrophobic organic compound partitioning [6,7,31,32,42]. The present study results suggested the sorption behavior and chemical properties in low molecular weight DOM should consider in investigating hydrophobic organic compound sorption on DOM. The fluorescence method is an effective method that the obtaining sorption constant and chemical properties of DOM. The DOM separation into size-fractioned DOM to investigate sorption constant of different molecular weight DOM is essential to understand BPA/OP's sorption behavior.

Comment #4-2

Solution composition. there is plenty of evidence to show impacts of other water quality paramters on the nature of DOM and also interaction during the sorption of organic compounds to DOM/OM. salinity, other ions (complexing/competing), pH etc. you have only one set of experimental conditions, these are not fully described, but the dynamic external environment should be considered

Response #4-2

Thank you for the comment and suggestion.

We agree with the impacts of other water quality parameters on the nature of DOM and interaction during the sorption of organic compounds to DOM/OM. In this study, the sorption behavior was conducted on the natural condition. We focused on the size-fraction and chemical properties of DOM that influences the sorption behavior between OP/BPA and size-fractioned DOM. Besides, the hydrophobicity of OP/BPA impact on sorption constant was also discussed. The water quality parameters on the nature of DOM and even interaction during the sorption of organic compounds to DOM/OM should be examined in future studies.

Comment #4-3

Above points will also impact on measurement conditions for fluorescence and UV VIS. Commentary on this also needed.

Response #4-3

Thank you for your comment and suggestion.

The Conclusion section has been significantly rewritten following the reviewer's suggestion. The parameters impact on measurement condition for fluorescence and UV-Vis also discussed in response #4-2.
